# Polarization-Sensitive Optical Coherence Tomography for Monitoring De- and Remineralization of Bovine Enamel In Vitro

**DOI:** 10.3390/diagnostics14040367

**Published:** 2024-02-07

**Authors:** Stella M. M. Hund, Jonas Golde, Florian Tetschke, Sabine Basche, Melina Meier, Lars Kirsten, Edmund Koch, Christian Hannig, Julia Walther

**Affiliations:** 1Department of Medical Physics and Biomedical Engineering, Faculty Carl Gustav Carus of Medicine, TU Dresden, Fetscherstraße 74, 01307 Dresden, Germany; stella_maria_margaretha.hund@tu-dresden.de (S.M.M.H.); jonas.golde@iws.fraunhofer.de (J.G.); lars.kirsten@tu-dresden.de (L.K.); 2Polyclinic of Operative Dentistry, Periodontology and Pediatric Dentistry, Faculty of Medicine Carl Gustav Carus, TU Dresden, Fetscherstraße 74, 01307 Dresden, Germany; florian.tetschke@sonovum.de (F.T.); sabine.basche@ukdd.de (S.B.); melina.meier@ukdd.de (M.M.); christian.hannig@ukdd.de (C.H.); 3Clinical Sensoring and Monitoring, Department of Anesthesiology and Intensive Care Medicine, Faculty of Medicine Carl Gustav Carus, TU Dresden, Fetscherstraße 74, 01307 Dresden, Germany; edmund.koch@tu-dresden.de

**Keywords:** optical coherence tomography, polarization-sensitive, degree of polarization, depolarization, diagnostic imaging, dental caries, dental enamel, tooth demineralization, tooth remineralization

## Abstract

Early caries diagnosis still challenges dentistry. Polarization-sensitive optical coherence tomography (PS-OCT) is promising to detect initial lesions non-invasively in depth-resolved cross-sectional visualization. PS-OCT with determined degree of polarization (DOP) imaging provides an intuitive demineralization contrast. The aim of this study is to evaluate the suitability of DOP-based PS-OCT imaging to monitor controlled de- and remineralization progression for the first time and to introduce it as a valid, non-destructive in vitro detection method. Twelve standardized bovine enamel specimens were divided in different groups and demineralized with hydrochloric acid (HCl) as well as partly remineralized with fluoride over a 14-day pH-cycling experiment. The specimens were stored in artificial saliva and sodium chloride (NaCl), respectively. Progress measurements with PS-OCT were made with polarization-sensitive en faceand B-scan mode for qualitative evaluation. The specimens demineralized in HCl showed the most pronounced surface change (lowest DOP) and the most significant increase in depolarization. Additional fluoride treatment and the storage in artificial saliva resulted in the opposite (highest DOP). Therefore, DOP-based PS-OCT imaging appears to be a valuable technique for visualization and monitoring of enamel demineralization and remineralization processes in vitro. However, these findings need to be confirmed in human teeth ex vivo or in situ.

## 1. Introduction

The detection of early carious lesions and monitoring its progression is still challenging in modern dentistry. Initial lesions are caused by demineralization of the tooth surface, which corresponds to a process in the enamel where calcium and phosphate precipitates without a substance defect. Dried, initially demineralized enamel appears whitish or opaque on visual examination, known as “white spots”, which cannot be detected with conventional X-ray [1]. So far, such a treatment decision can only be made subjectively based on dentists’ experience and the color and the shape of the lesion [2]. These lesions can be reversed by remineralization, i.e., the re-deposition of minerals. De- and remineralization are continuous and dynamic processes, thereby predominating remineralization prevents caries progression followed by cavitation [3,4]. To avoid invasive therapeutic intervention, it is important to detect initial caries early and monitor it closely. Dietary changes, antibacterial therapy and fluoride application can stop or remineralize these non-cavitated lesions [5]. Gomez and Al Saffan discussed some currently available non-invasive methods, such as quantitative light-induced fluorescence, DIAGNOdent, fiber-optic transillumination and electrical conductance for detection of early caries [6,7] with the result that all caries detection methods alone are subject to error and that additional methods should be used to aid clinical decision-making. In addition, Intraoral scanners using near-infrared transillumination or light-induced fluorescence technology are among the latest innovations for X-ray-free caries diagnostics [8,9]. They offer a fast in real-time caries imaging which is patient-friendly [10].Further research is needed to improve the accuracy of these scanners, as non-cavitated lesions have often been misclassified as healthy [10,11]. A promising non-invasive diagnostic imaging technique is polarization-sensitive OCT (PS-OCT), which is based on polarization change in the backscattering light [12,13]. OCT in general allows for in depth-resolved 2D and 3D cross-sectional imaging of biological tissues in real time without ionizing radiation [14,15], and can be used to observe enamel alterations longitudinally [16,17,18,19,20].

In order to use PS-OCT clinically and to interpret the images correctly, basic research is essential. Numerous models have been developed that induce caries-like surface changes under constant conditions in enamel specimens [21,22]. The knowledge of the effect of fluoride is based on in vitro pH-cycling studies [23]. Bovine teeth in particular are well suited for remineralization studies, as they have not undergone fluoride therapy and have similar properties to human teeth [24,25]. The advantage of PS-OCT in pH-cycling studies is the non-destructive follow-up monitoring of tooth specimens. In the past, initial changes in the enamel have been visualized in detail only histologically. In vitro OCT studies [26,27] also compare mineral loss and lesion depth using digital microradiography or transverse microradiography [28] as a reference, since this is currently considered as the gold standard of early caries detection. The disadvantage of comparing histological techniques is that samples from the experiment have to be destroyed. Thus, further lesion progression of the same specimen cannot be demonstrated.

Most of the published PS-OCT studies used intensity-based imaging, but there are also a few studies using polarized light, which has been proven to detect the different stages of artificially demineralized enamel [29]. A qualitative evaluation of lesion progression for real-time diagnosis with optical polarimetric detection of dental hard tissue diseases was made by Hsiao et al. [30] and provided promising diattenuation and retardation images.

In this study, depolarization imaging by means of calculating the degree of polarization (DOP) of PS-OCT data sets is applied for the first time in a pH-cycling model with fluoride treatment. This allows the analysis of the surface area of the enamel slabs in longitudinal direction. DOP was defined in ophthalmology as a tissue-specific contrast for the segmented retinal epithelium [31], and was first used in dentistry by Golde et al. [32,33]. Currently, depolarization imaging using DOP by PS-OCT non-destructively is considered the most suitable method for detecting early and advanced caries lesions due to significant contrast and ease of interpretation [32,34,35], for which reason it is chosen in the following study. The aim is to establish PS-OCT with DOP based imaging as a valid alternative to destructive detection methods in vitro and thus provide a basis for perspective clinical application.

## 2. Materials and Methods

For the cyclic in vitro investigation of de- and remineralization processes in the enamel specimen model, the following study design with appropriate sample preparation was chosen and is described in more detail in the following sections.

### 2.1. Specimens’ Preparation and Measuring Sequence

An overview of the methodology is shown in Figure 1, and includes the following steps in detail:

First, 48 bovine incisors of 2-year-old cattle were prepared under water cooling to specimens with a diameter of 5 mm, polished (grit size 1200) and cleaned with sodium hypochlorite (3%) according to an established protocol in order to remove the smear layer [34,36]. This resulted in a leveled, polished enamel side of the bovine specimens and a leveled dentin back side, with slight differences in the thickness of the enamel and dentin, respectively. Subsequently, all 48 specimens were degreased with 100% ethanol for 1 min in an ultrasonic bath. Nail polish (gel nailpolish red 117, Cosnova GmbH, Sulzbach, Germany) [37] was applied thinly with a synthetic fiber brush (da Vinci Artist brush factory DEFET GmbH, Nuremberg, Germany) halfway down the front of the specimens and allowed to air dry at room temperature for approximately 30 min.

Secondly, the enamel specimens were then distributed in eight groups of six specimens each in a randomized manner and labeled with nail polish. Each group (I–VIII) was further divided into three subgroups (A, B, C), each containing two specimens used for three different measurement sequences for PS-OCT:(A)Specimens 1, 2: Follow-up (PS-OCT on day 0, 3, 5, 7, 10, 12, 14).(B)Specimens 3, 4: Removal after 7 days (PS-OCT on day 0 and 7).(C)Specimens 5, 6: Follower (PS-OCT on day 0, 7, 14).

Third, all specimens were imaged by PS-OCT in accordance to the measurement sequence of each subgroup as listed above within the 14-day-study-period, and additionally, in baseline condition once before and after nail polish was applied.

### 2.2. Liquid Preparation

For demineralization, hydrochlorid acid (HCl, pH 2.7) was prepared, and the pH-value was measured every 3 days and adjusted if necessary. The fluoride for remineralization used was Elmex Professional Opti-Enamel (pH 4.5) (Sealing and Strengthening Mouthwash, CP GABA GmbH, Hamburg, Germany). It contains amine fluoride and sodium fluoride (500 ppm fluoride) and stannous chloride (800 ppm stannous). For every cycle, fresh solutions were used.

As storage solutions, self-prepared physiological saline solution (0.9% NaCl) or artificial saliva were used, which were both renewed every day. The artificial saliva solution (see Section A.3) was a ready-mix from the clinic pharmacy of the University Hospital Carl Gustav Carus and was stored at below 25 °C in the fridge.

### 2.3. Artificial Demineralization and Remineralization pH-Cycling Model

The experimental study on enamel slabs was conducted as a cyclic model over 14 days as shown in Figure 2. The scheme was loosely based on [38,39], and performed as follows:

**Step 1:** at the beginning of each cycle, the well plates were filled with 8 mL of the respective liquids for each group (I–VIII) using serological pipets (Stripettes, Eppendorf SE, Hamburg, Germany). **Step 2:** In the morning (8 a.m.), at noon (12 p.m.) and in the evening (6 p.m.), the demineralization procedure was performed. A demineralization cycle lasted 10 min and consisted of five repetitions of 1 min HCl, followed by 1 min Storage Solution (NaCl or artificial saliva depending on the group). The aim was to create an in vivo-like situation in which the teeth in the mouth are rinsed by saliva after exposure to acid through eating and drinking. **Step 3:** The remineralization process with fluoride was applied in the morning and evening directly after demineralization. For this purpose, the specimens were placed in the fluoride solution for 2 min and then transferred to the well plates for storage. To create similar conditions as in vivo, the cyclic process was applied in a water bath and the well plates were stored at 37 °C in an incubator with gentle agitation during the residual periods. To prevent evaporation of the liquid, each well plate was covered with a plastic sheet. As shown in Figure 1, PS-OCT measurement was performed regularly every 2 to 3 days for subgroups A, and every seven days for subgroup B and C, respectively. After 14 days, the nail polish on all specimens was removed with pure acetone [40]. To prevent alterations due to dehydration by acetone, the specimens were stored in distilled water for one day before the final PS-OCT measurement was performed, for purpose of re-hydration.

### 2.4. Polarization-Sensitive OCT Imaging

A commercial benchtop spectrometer-based PS-OCT system (Telesto^®^ Series, Model: TEL220PS, Thorlabs GmbH, Luebeck, Germany) with a central wavelength of 1300 nm and a spectral range of over 170 nm was used to measure demineralization and remineralization on the bovine enamel plate model. The axial resolution resulting from the spectrometer configuration is 5.5 µm in air and the axial measurement range is 3.5 mm in air. For the in vitro studies, the spectral domain PS-OCT system was operated at an A-scan rate of 28 kHz with a sensitivity of over 100 dB according to the manufacturer. For high-resolution imaging in the lateral direction (orthogonal to the incident sample beam), the system was combined with an objective lens (LSM02, Thorlabs GmbH, Luebeck, Germany), which allows telecentric imaging with a lateral resolution of 7 µm (1/e^2^ beam diameter at focus) at a working distance of 7.5 mm and a field of view of 6 × 6 mm^2^.

Polarization-sensitive imaging is allowed by illuminating the sample with a single circular input state. The light reflected from the sample, and changed in polarization if birefringence is present, is superimposed with diagonally polarized reference light and detected by means of two spectrometers for the detection of the orthogonal linearly polarized states (co- and cross-polarization). Using the interferograms of the horizontally and vertically polarized components of the back-reflected light, the so-called reflectivity depth scans are calculated from the complex-valued depth signals (A-scans) of the co- and cross-polarization channel by summing up the squares of the amplitudes [32]. Moreover, DOP can be determined as a measure of the depth-resolved enamel demineralization [35]. For DOP calculation proposed in [32,35], the measured Jones vectors are converted into Stokes components. For contrast-enhanced depolarization imaging, the so-called noise-immune DOP approach proposed by Makita et al. [41] was implemented. Finally, the reflectivity information is multiplied to the colored noise-immune DOP images to combine the structural with the depolarization information [32,35].

For in vitro imaging of de- and remineralization over time, the prepared enamel specimens were fixed on a microscope slide and imaged in 3D with PS-OCT at an oblique position to the orthogonal plane of the incident sample beam (see Figure 3c). To avoid hydration-related variations in scattering within the enamel [42], the platelets were imaged with PS-OCT immediately after removal from the respective solution. With a measurement time of 30 s for the 3D PS-OCT measurement, dehydration of the enamel specimens has been avoided. By means of the detected interferograms of the co- and cross-polarized channel of the detected PS-OCT volume stacks, both reflectivity and DOP cross-sectional images through the center of the enamel specimens, as well as corresponding en face visualization were generated (see Figure 3b). Edge detection of the 3D reflectivity scans was used to level of the surface of the enamel specimens (see Figure 3d) and serves for the subsequent artifact-free en face DOP representation (see Figure 3e), which was determined by averaging the DOP over a depth range of 30 pixels (65 µm at a refractive index of 1.631 of enamel for the central wavelength of 1300 nm [43]).

## 3. Results

The depiction of the PS-OCT DOP en face and corresponding surface-levelled B-scans images of one platelet of each group is shown in Figure 4. For all four groups on day 0, sound enamel is visible, which was covered with nail varnish on the left side. Thereby, sound enamel does not show depolarization, it appears blue due to the high DOP values. In groups II–IV, a slight increase in depolarization can be observed on day 7, which is noticeable by the increase in magenta (lower DOP value) on the right side of the enamel slabs in comparison to day 0. The nail polish itself had a higher depolarizing effect, as shown by the strongly reduced DOP in the PS-OCT cross-sectional and en face DOP images. The application thickness or a reaction with the liquids (respectively the acid HCl) may play a role for that phenomenon of the NP. On day 14, the difference between the groups with (I and III) and without (II and IV) fluoride treatment is clearly distinguishable. Depolarization has decreased in groups I and III from day 7 to day 14. In group II and IV, a further increase in depolarization is noticeable, which is strongest in group IV (storage in NaCl). This becomes evident in the subsurface layer of the enlarged views.

An overview of the DOP en face projections of the follow-up platelet measurement over several days of subgroup A (dense follow up c.p. Table 1) of each group I–IV is shown in Figure 5. During the course of the experiment from day 0 to day 14, the color-coded DOP decreases and changes more and more to magenta in group II and IV due to increased demineralization combined with an increased light scattering [32,34,35]. By means of the en face DOP projections, depolarization in group II appears less pronounced compared with group IV. However, in agreement of groups II and IV, demineralization develops locally and irregularly and appears inhomogeneous in the en face DOP images over the time course of the study. The demineralization on day 14 in group IV is finally more homogeneous and more pronounced compared to group II.

Group III treated with fluoride initially shows an increase in demineralization until day 7 and then a decrease until day 14. This can be interpreted as remineralization. In group I, there is a marginal DOP decrease up to day 7, followed by an DOP increase on the following days. The black box on the right side of Figure 5 presents the corresponding enamel specimens after the nail polish has been removed. The corresponding photographs and DOP surface-leveled center B-scans of the platelets after NP removal (black box in Figure 5) are presented in Figure 6.

After removal of the nail polish (NP) with acetone, a clear demarcation of the exposed area (right side) from the unexposed area (left side) becomes visible. This is apparent in the case of group II–IV. In group IV, the exposed surface shows the strongest demineralization by means of the increased DOP (magenta) and the macroscopic enamel translucency has become opaque, whitish and rough (Figure 6 lower right). The light scattering has increased and the EDJ in the larger depth is not identifiable. Considering the visual impression by photographs, in group II, the white spot lesion on the right side of the specimen is similar to group IV, whereas the depolarization on the B-scan is not as strong and more punctual. The right specimen’s side on the photographs of group I and III is more translucent and appears slightly darker than the left one originally covered with NP. No white spot lesion has occurred due to fluoride treatment, which seems to have prevented the demineralization. In group I, there was hardly any surface change. The combination of artificial saliva and fluoride seems to have protected the enamel surface and has the best balance between de- and remineralization. Comparing the storage solutions NaCl and artificial saliva, the buffering properties in the artificial saliva seem to protect the enamel from demineralization or it might have a remineralizing effect. As expected, storage of the specimens in HCl and NaCl results in the strongest surface alterations and the lowest DOP.

In comparison, the control groups V-VIII show little or no change in and below the enamel surface (see Appendix Figure A2 and Figure A6). These specimens did not receive any acid attack by HCl, resulting in no mineral loss to be reversed by remineralization.

## 4. Discussion

It has been shown for the first time that depolarization imaging by PS-OCT is able to non-destructively visualize the process of early de- and remineralization. The different stages can be measured regularly, and the development and progression of the lesions can be easily assessed and interpreted in 3D-scans. Minimal surface alterations of the specimens can be clearly read qualitatively and intuitively in PS-OCT-based depolarization mode. In our opinion, the quick and straightforward interpretation makes depolarization imaging by PS-OCT the method of choice for non-destructive treatment monitoring in in vitro and further in situ studies [44,45].

Many studies use polarized light microscopy (PLM) or transverse microradiography (TMR) as a reference for lesion assessment in PS-OCT in vitro studies [28,46,47,48,49], both of which are destructive methods due to the thin sections required. The advantage of PLM is that histological features such as the differentiation of characteristic zones of demineralization or remineralization can be examined and the lesion depth can be measured [34]. Due to the numerous comparative studies [28,34,46,47,48,49,50], all of which found a significant correlation between lesion depth in PS-OCT and PLM, no histological sections were considered necessary and consequently made in this study. TMR can be used to determine the percentage mineral content in healthy, demineralized and remineralized areas. The statistical analyses of mineral loss and lesion depth in the different stages of early demineralization and remineralization could supplement the qualitative evaluation of PS-OCT DOP images in our study.

A non-destructive complementary in vitro method for early caries detection is X-ray micro-computed tomography (micro-CT), which can determine mineral density and mineral loss [51,52,53]. Micro-CT scans are very time-consuming, the high-resolution 3D-images of micro-CT are comparable [51], as are the depolarization signals, which are in a good correlation with PS-OCT measurements [52], thus it has a low additional value to PS-OCT. Due to the numerous previous comparative studies that concluded that PS-OCT is well suited for the assessment of caries lesions [49,54] and its inhibition by anti-caries agents [55], we did not make a comparison in the presented study. In situ applications with PS-OCT are, so far, only possible to a limited extent. A further study is conceivable in which enamel specimens are fixed in individual splints and worn in situ [45,56,57,58]. By simply removing and reinserting the splint, PS-OCT measurements can be performed quickly in vitro and de- and remineralization in the oral cavity can be monitored.

PS-OCT imaging is very promising, especially for the detection of early alterations on enamel surfaces. Numerous studies confirmed that demineralization results in the formation of a subsurface layer with intense optical backscattering [48]. Enamel is highly birefringent, thus demineralization leads to a significant increase in light scattering and large changes in birefringence [59]. It is also proven that the intensity of linearly polarized light scattered into the perpendicular polarization state (depolarized) can be directly correlated with the degree of demineralization and lesion severity [48]. The increase in depolarization in PS-OCT DOP-based images of group II and IV due to demineralization is consistent with results in the literature [22,32,34].

Consequently, DOP-based PS-OCT imaging is well qualified for demineralization processes, and should provide equally promising results for remineralization processes in the pH-cycling model. Since this hypothesis cannot yet be validly tested in vivo for various reasons (e.g., patient individuality, probe technique, etc.), the pH-cycling model is used in vitro. Regarding the specimens of the presented study, bovine teeth are very homogeneous in age, nutritional status and fluoridation measures, thus allowing more uniform de- and remineralization compared to human teeth [60]. To minimize sample-specific variabilities (microcracks, surface roughness), the progress of the same samples during the entire protocol was investigated. According to Mellberg, bovine teeth have a more porous structure compared to human teeth, therefore the diffusion rate is higher and carious lesions appear faster [56,61]. However, by flattening and polishing the surface in the preparation process, a lower susceptibility to de- and remineralization is expected compared to clinical conditions [62]. With regard to the used nail polish for generating an internal enamel reference during the pH-cycling procedure, it was seen that the surface of the NP changed its polarization during the experiment. Furthermore, few bubbles have appeared in the NP during the storage in the liquids. According to Cara et al. [63] and Kang et al. [64], who used red nail varnish in their OCT caries studies, nail polish is acid resistant. Removing the nail polish with acetone could have changed the surface despite subsequent dehydration. Some red color particles remained in cracks (see the photograph on day 14 in Figure 3). Bovine teeth are particularly suitable for this study, as they provide results with greater consistency in a coherent study [65].

The complex experimental procedure of the proposed study, with constant conditions and mimicking oral pH-fluctuations, was chosen to demonstrate the suitability of DOP imaging for monitoring the initial surface change due to intercalation of minerals. Daily measurement of all specimens was not possible for organizational reasons and due to the tight cycle schedule. Artificial saliva, unlike human saliva, is not subject to daily fluctuations and does not contain any proteins. Salivary proteins can facilitate or aggravate early de- and remineralization [44]. This means that in this study, no pellicle layer can be formed on the surface which protects the enamel from the influence of various acids and from calcium loss [24]. In order to reproduce the oral situation correctly, a follow-up study in situ or with collected human saliva in vitro is necessary. Due to the buffering properties and associated remineralization capacity of artificial saliva [66,67], there is a reduction in enamel erosion, which can be seen in the DOP images of group I and II in comparison to groups in NaCl.

Fluoride promotes remineralization, inhibits demineralization and forms a protective calcium fluoride top layer [68]. The mouth rinse used in this study contains stannous chloride, which forms very acid-resistant precipitates on the tooth surface [57,58]. Mineral deposition during remineralization is visible in the outer layer on PS-OCT [40,54]. According to Jones et al., remineralization of severe early artificial caries can be quantified by measuring an increase in reflectivity in the vertical axis without an accompanying change in lesion depth. In very early artificial lesions, PS-OCT reveals a highly scattering and depolarizing outer layer after fluoride-containing remineralization [60]. In the study presented here, the inhibition of the initial caries lesions could be successfully measured by PS-OCT DOP representations, which is confirmed by the high DOP values in the results. Chong et al. [55] were also able to visibly demonstrate an inhibition of artificial demineralization by fluoride through significantly lower reflection in the PS-OCT measurements. Although the results of the PS-OCT images in the study by Fatemah Memon et al. for the anti-caries treatment were probably ambiguous due to intensity changes in the light scattering properties, it appears that reduced light scattering occurred with a reduction in porosity during mineral precipitation during remineralization [3]. Fetemah Memon et al. described remineralization as the process of calcium, phosphate, and fluoride becoming disorderly deposited in voids (porosities), resulting in reduced backscattering [3]. The primary effect of the acid is an increase in porosity, which results in an increase in backscattering and a decrease in depolarization. Consequently, a decrease in porosity causes a decrease in scattering and an increase in depolarization due to intercalation of the minerals. This could be confirmed in our study, with the clear results of the PS-OCT depolarization progression images by the increasing DOP values in the groups with fluoride treatment (group I and III). In vivo, this process is modulated by pellicle formation and penetration of proteins into the demineralized lesions. This has to be considered in further in situ studies [44]. Within the limitations of this in vitro study, we propose a quantitative evaluation of the PS-OCT DOP images to achieve greater objectivity of the results. Until now, the detection of caries lesions of human teeth using PS-OCT-DOP imaging has already been evaluated by Golde et al. [32,69]. Further research, in particular to investigate early de- and remineralization processes of human teeth, are still needed, both ex vivo and in vivo. For a prospectively clinical use, a handheld OCT [70,71] is necessary.

## 5. Conclusions

In summary, DOP-based PS-OCT imaging has performed as a valid detection and visualization method for initial caries lesions and for monitoring de- and remineralization processes. The en face and B-scan projections were robust and easy to interpret visually. Within the limitations of thisin vitro study using bovine teeth, this technology has great potential for non-destructive analytics and, prospectively, for clinical applications. Follow-up ex vivo studies on human teeth are required to generate the necessary expertise for a clinical pilot study.

## Figures and Tables

**Figure 1 diagnostics-14-00367-f001:**
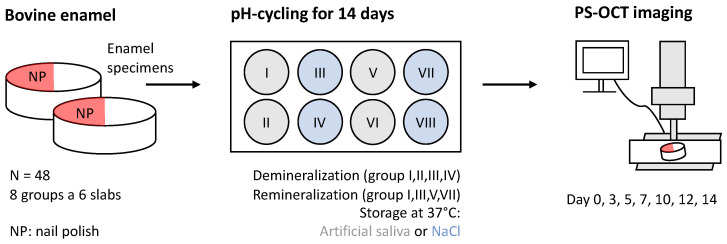
Workflow of the artificial de- and remineralization process.

**Figure 2 diagnostics-14-00367-f002:**
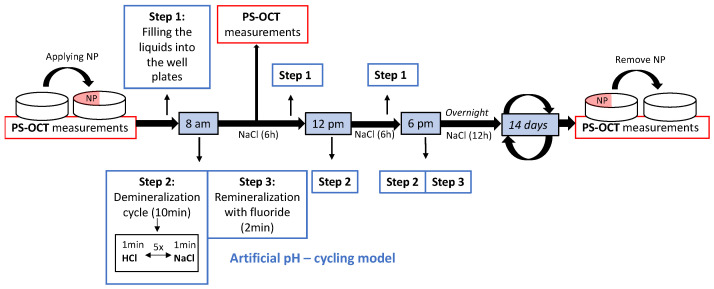
Timetable for incubation of specimens from group I. The schedule for group II would be the same, except for step 3, as no fluoride treatment was performed. For group III, the storage solution was NaCl, so artificial saliva was replaced by NaCl (Table 1).

**Figure 3 diagnostics-14-00367-f003:**
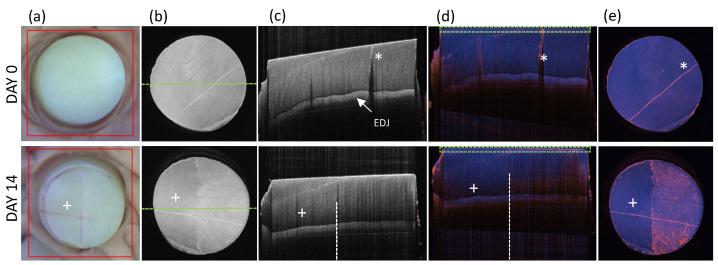
Illustration of the different projections of one representative enamel specimen of group II, platelet 2 day 0 before the application of the nail polish (NP) and on day 14 after its removal. (**a**) Photographs of the enamel platelets served by the integrated VIS camera of the commercial PS-OCT system marking the region of the PS-OCT volume scan by the red box. (**b**) PS-OCT en face projections by means of the surface-leveled reflectivity 3D stacks. The green dashed line through the center of the slab marks the position of the PS-OCT cross sectional image plane. (**c**) PS-OCT reflectivity cross-sections with obliquely detected enamel slab surface; (**d**) PS-OCT straightened DOP cross-sections. The green box illustrates the depth range of 30 pixels (65 µm at a refractive index of 1.631 for 1300 nm central wavelength) used for the PS-OCT DOP en face projection; (**e**) PS-OCT DOP en face projection. EDJ: enamel dentin junction. Asterisk: micro-crack. Plus sign: NP protected side after NP removal.

**Figure 4 diagnostics-14-00367-f004:**
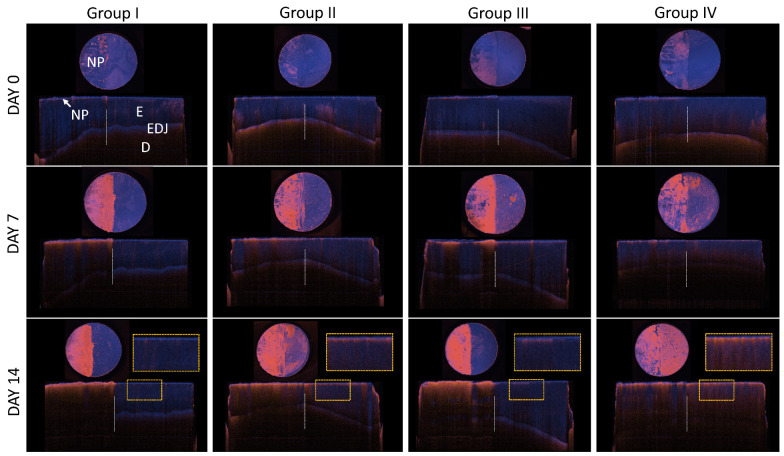
Overview of PS-OCT DOP en face projections and straightened DOP center B-scans at day 0, day 7 and day 14 (Table 1). In the enlarged views (yellow dashed frame), the different extent of depolarization of the subsurface layer becomes visible. E: enamel, EDJ: enamel dentin junction, D: dentin, NP: nail polish; Group I, III, IV: specimen 1, Group II: specimen 6.

**Figure 5 diagnostics-14-00367-f005:**
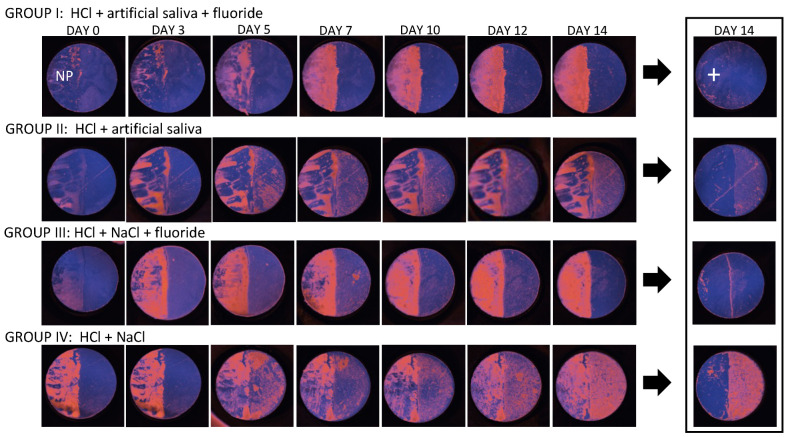
Overview of the en face DOP projections by means of the PS-OCT measurements of group I–IV (subgroup A) with nail polish (NP) and after NP removal. Group I–IV: enamel specimen 1. NP: nail polish. Plus sign: NP protected side after NP removal.

**Figure 6 diagnostics-14-00367-f006:**
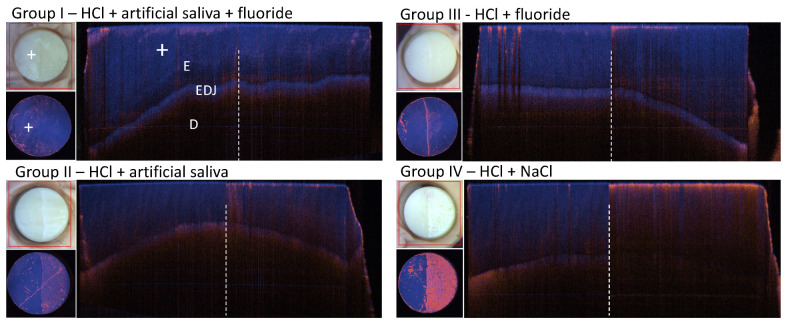
PS-OCT results on day 14 after nail polish removal: Photograph, PS-OCT DOP en face and the corresponding surface-leveled center B-scans after removal of NP. On the B-scans and en face projections, the left slab sides were protected by NP during the pH-cycling procedure and, as expected, no surface change was observed in contrast to the right enamel slab sides, which were exposed to the solutions. Group I–IV: enamel specimen 1. E: enamel, D: dentin, EDJ: enamel dentin junction, Plus sign: NP protected side after NP removal.

**Table 1 diagnostics-14-00367-t001:** Specimen division: I–IV experimental groups (each with *N* = 6 specimens for subgroups A, B, C), V–VIII control groups (each with *N* = 6 specimens for subgroups A, B, C).

Group Nb	Storage	Demineralization	Remineralization
I	Artificial Saliva	HCl	Fluoride
II			-
III	NaCl		Fluoride
IV			-
V	Artificial Saliva	-	Fluoride
VI			-
VII	NaCl		Fluoride
VIII			-

## Data Availability

The data presented in this study are available on request from the corresponding author.

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
