# Peer review of "Polarization-Sensitive Optical Coherence Tomography for Monitoring De- and Remineralization of Bovine Enamel In Vitro"

_diagnostics, 2024, doi:10.3390/diagnostics14040367_

Round 1

Reviewer 1 Report

Comments and Suggestions for Authors

With interest I’ve read the paper “Progress monitoring of enamel de- and remineralization in vitro by polarization-sensitive optical coherence tomography”. The authors conducted an in vitro study to evaluate the suitability of degree of polarization-based optical coherence tomography imaging to monitor experimental de- and remineralization process. Early caries detection with the use of non-invasive diagnostic methods is an important and interesting topic. The topic is relevant to a range of readers among conservative dentists and researchers.

I would suggest to rephrase the title to: the use of polarization-sensitive optical coherence tomography for monitoring enamel de- and remineralization in vitro 

General note: the study is well-written and thoroughly planned. All sections of the manuscript contain essential details.

Comments:

  1. Lines 284-288 “The advantage of a pH-cycling model is that it mimics the dynamics of mineral loss and gain during caries formation in the oral cavity [58]. The pH-value of hydrochloric acid (pH 2.7) was chosen since soft drinks lie in this range, thus a good reference to the real situation can be established [30,59]. The twice-daily application of fluoride illustrates the brushing of teeth in the morning and in the evening.”

This part rises questions. First of all, the “natural” caries demineralisation process is caused by lactic acid produced by bacteria. This acid is soon removed or neutralised by saliva and remineralisation of the superficial layer occurs. Therefore, the natural white spot caries presents as a SUBsurface demineralisation. The ph may by much higher than 2,7 (around 4-5). On the other hand, the intake of acidic soft drinks can demineralise enamel causing acid erosion. The authors need to avoid confusion between erosion and caries processes.

  2. Why did not the authors try to do some quantitative analysis? I think, programmed assessing of the color changes could provide more objectivity for the result.

  1. Mentioning the limitations of the study and suggestions for further research are advisable at the end of the discussion section

Despite the comments, the paper is well-written and interesting and can be published after minor revision.

Author Response

We would like to thank the reviewer for his time and effort to review and assess our submitted manuscript and have conscientiously taken their comments into account in our revision.

Please see the attachment file.

Reviewer 2 Report

Comments and Suggestions for Authors

Figure 1 caption stated, "For group III, the storage solution was artificial saliva, so NaCl was replaced by artificial saliva (Table 1)."

Nevertheless, in Figure 5  NaCl is  used for group III and similar for Figure A3

Author Response

We would like to thank the reviewer for his time and effort to review and assess our submitted manuscript.

Reviewer 3 Report

Comments and Suggestions for Authors

Dear authors,

I really appreciate your effort on this research topic.

The general approach is interesting and from my point of OCT is a promising technology for application in dentistry. The manuscript is well written and in good order. For better clearance, I suggest to address the following points:

1) Title: Please add to the title that you investigated bovine enamel specimens.

    Keywords: Is suggest to use MESH terms as keywords. Please clarify that you investigated bovine teeth.

2) Introduction:

  • Line 23 is a double space before “Dried”#
  • In addition to Diagnodent or Diagnocam, some intraoral scanners allow caries diagnosis. Data are scarce, but this is another technology that should be mentioned when talking about recent innovations in caries diagnostics, even if they are not good at detecting early lesions. This highlights the value of OCT technology.

3) Materials & methods:

  • Please explain the number of specimens.
  • The distribution of the groups is not clear for me. Please revise the chapter 2.1 with the Table 1.
  • Was the OCT used a desktop OCT? Please add missing information.
  • Line 142 is a double space before “The light”
  • DOP is already explained in the Introduction section.
  • Line 164 is a double space before “Edge”
  • I was wondering why you did not check your results from the last probe with histological sections. This is a common way to verify results with established methods.

4) Discussion:

·       The Discussion section is very long. Please shorten the Materials & Methods discussion once your study design regarding the specimens is established.

·       Please discuss why you did not control your results with histological sections at the end.

·       Please discuss that you need a handheld OCT for in vivo application. Although the measurement took only 30 s, this can be a long time in vivo, especially for a handheld. Please discuss your ideas on this topic.

5) Conclusion (in Abstract and main text):

Since you studied bovine teeth, please rephrase the conclusion section regarding this point. As you have correctly discussed, the results of your work need to be proven in extracted human teeth before you can write this conclusion. But you are one step behind, so please rephrase the sentence in the abstract: "Therefore, DOP-based PS-OCT imaging appears to be a valuable technique for visualizing and monitoring enamel demineralization and remineralization processes in vitro. However, these findings need to be confirmed in human teeth ex vivo and in situ".

In the conclusion section: "Within the limitations of this in vitro study using bovine teeth...". Please include human teeth in the ex vivo studies for clarity.

6) Literature

Please update the literature. Of the 64 references, there is only one reference from 2023, two from 2022, one from 2021, and two from 2020.

Good luck and keep well!

Author Response

We would like to thank the reviewer for his time and effort to review and assess our submitted manuscript and have conscientiously taken their comments into account in our revision.

Round 2

Reviewer 3 Report

Comments and Suggestions for Authors

Dear Authors,

Thank you for the revision of your manuscript. All points have been adequately addressed.

Good luck and keep up the good work!